# CSConv2d: A 2-D Structural Convolution Neural Network with a Channel and Spatial Attention Mechanism for Protein-Ligand Binding Affinity Prediction

**DOI:** 10.3390/biom11050643

**Published:** 2021-04-27

**Authors:** Xun Wang, Dayan Liu, Jinfu Zhu, Alfonso Rodriguez-Paton, Tao Song

**Affiliations:** 1College of Computer Science and Technology, China University of Petroleum, Qingdao 266580, China; wangsyun@upc.edu.cn (X.W.); z19070030@s.upc.edu.cn (D.L.); 2High Performance Computer Research Center, Institute of Computer Technology, Chinese Academy of Sciences, Beijing 100190, China; 3School of Economics, Beijing Technology and Business University, Beijing 100048, China; 4Department of Artificial Intelligence, Faculty of Computer Science, Polytechnical University of Madrid, Campus de Montegancedo, 28660 Madrid, Spain; arpaton@fi.upm.es

**Keywords:** protein-ligand binding affinity, 2-D structural CNN, spatial attention mechanism

## Abstract

The binding affinity of small molecules to receptor proteins is essential to drug discovery and drug repositioning. Chemical methods are often time-consuming and costly, and models for calculating the binding affinity are imperative. In this study, we propose a novel deep learning method, namely CSConv2d, for protein-ligand interactions’ prediction. The proposed method is improved by a DEEPScreen model using 2-D structural representations of compounds as input. Furthermore, a channel and spatial attention mechanism (CS) is added in feature abstractions. Data experiments conducted on ChEMBLv23 datasets show that CSConv2d performs better than the original DEEPScreen model in predicting protein-ligand binding affinity, as well as some state-of-the-art DTIs (drug-target interactions) prediction methods including DeepConv-DTI, CPI-Prediction, CPI-Prediction+CS, DeepGS and DeepGS+CS. In practice, the docking results of protein (PDB ID: 5ceo) and ligand (Chemical ID: 50D) and a series of kinase inhibitors are operated to verify the robustness.

## 1. Introduction

Nowadays, there are a large number of drug candidate compounds in large databases such as ChEMBL [1] and PubChem [2]. The binding affinity of small molecules to receptor proteins is the key to drug discovery and drug repositioning [3]. Usually, chemical prediction methods are time-consuming and costly. Out of the 20,000 proteins in the human proteome, less than 3000 of them are targeted by known drugs. Simultaneously, in order to reduce the cost of drug development and the risk in the process of drug research, it has become an important strategy to repurpose the approved drugs and explore their new functions. Nevertheless, due to the huge amount of data and high-throughput screening (HTS), the discovery process of novel drugs is expensive and time-consuming [4].

The development of accurate prediction models for calculating binding affinity is imperative. Machine learning models are used to predict the interaction between drugs and targets on the basis of their chemical and biological characteristics, such as Support Vector Machine [5], Random Forest [6], deep learning methods [7,8,9,10] and so on. These major breakthroughs in deep learning have also pushed machine learning closer to the original human goal of “artificial intelligence”. The main advantage of deep learning is that it can automatically learn data features using a common algorithm structure framework. This framework is usually composed of a stack of multilayer simple neural networks with nonlinear input and output mapping characteristics, including deep neural networks (DNN), deep belief networks (DBN) [11], convolutional neural networks (CNN), auto-encoders and recurrent neural networks (RNN).

Until now, plenty of deep learning frameworks and tools have been developed for various purposes in computational chemistry-based drug discovery, such as DeepDTA Drug-Target Affinity prediction [12]; GraphDTA using molecular graphs as the input of graph convolutional neural network [13]; and DeepPurpose integrating a variety of encoding methods of drug molecules and protein amino acid sequences for DTI prediction [14]. DeepGS inputs the sequence information and two-dimensional structure information of drug molecules as well as the protein sequence information into the model for prediction, while GraphDTA takes molecular graphs as the input of the model [15].

In this study, we propose a novel deep learning method for drug-target interactions (DTIs) prediction. The proposed model is named CSConv2d, which is improved from the original DEEPScreen model. It has a convolutional block attention module (CBAM) via the use of 2-D structural representations of compounds as the input instead of sequential features such as SMILES or molecular fingerprints [16]. Additionally, it is an added channel and spatial attention mechanism (CS) separate from the network structure. Functionally, the CBAM module can increase the nonlinear expression ability of the network and learn complex features from two-dimensional structure diagrams. Our CSConv2d can train a model for each target, and each model is independently optimized to accurately predict interacting small molecule ligands for a unique target protein. The main advantage of the module is increasing DTI prediction performances with the use of 2-D compound images instead of using conventional sequential features such as the molecular fingerprints and SMILES [17].

Data experiments are conducted on ChEMBLv23 datasets. The simulation results show that, in predicting protein-ligand binding affinity, our CSConv2d performs better than the original DEEPScreen model, as well as some state-of-the-art DTI prediction methods including DeepConv-DTI, CPI-Prediction, CPI-Prediction+CS, DeepGS and DeepGS+CS. In practice, the docking results of protein (PDB ID: 5ceo) and ligand (Chemical ID: 50D) are operated to verify the robustness.

## 2. Materials and Methods

### 2.1. Dataset

The ChEMBL database (v23) is employed to create the training dataset. ChEMBLv23 is a large, open-access drug discovery database that aims to capture Medicinal Chemistry data and knowledge across the pharmaceutical research and development process; it contains therapeutic targets and indications of clinical trial drugs and approved drugs, with a total of 1,961,462 different compounds and 13,382 targets.

In order to build a reliable training dataset, the data is filtered and preprocessed according to different types and bioactivity measurements (Figure 1). First, the data is filtered according to different attributes, such as “target type”, “taxonomy”, “assay type” and “standard type”. Then, if multiple measurements for a ligand-receptor data point were present, the median value was chosen and duplicates were removed. For the “assay type”, the functional assays were eliminated and only the binding assays were kept. Simultaneously, bioactivity measurements without the pCHEMBL value were removed; this value allows a number of roughly comparable measures of the half-maximal response concentration/potency/affinity to be compared on a negative logarithmic scale. After processing, the number of bioactivity points was 769,935. Bioactivities were extracted for activity values (IC50/EC50/Ki/Kd) of 10 μM or lower, with a CONFIDENCE_SCORE of 5 or greater for ‘binding’ or ‘functional’ human protein assays. For each target, compounds with bioactivity values ≤ 10 μM are selected as positive training samples and compounds with bioactivity values ≥ 20 μM are selected as negative samples. The 10 μM cutoff for activity specified here is in accordance with the method employed in the study of Koutsoukas et al. [18], representing both marginally and highly active compounds. The number of positive samples of each target is greater than the number of negative samples. To balance it, a target similarity-based inactive dataset enrichment method is applied to populate the negative training sets to make the number of negative samples equal to the number of positive samples. The idea of this method is that similar targets have similar actives and inactives. The positive and negative sample ratio of the finalized training dataset for 800 target proteins is close to 1:1, which is 475,238 and 430,590, respectively.

### 2.2. Attention Module

The CBAM is a lightweight and general module, which can be seamlessly integrated into any CNN architecture with negligible overheads and which is end-to-end trainable along with base CNNs. The module pay more attention to the object itself and thus has a better performance and interpretability. There are two independent submodules in CBAM, which are the channel attention module (CAM) and spatial attention module (SAM), as shown in Figure 2.

Since each feature map is equivalent to capturing a certain feature, channel attention helps to screen out meaningful features, that is, to “tell” the network which part of the original image features are meaningful. A pixel in the feature map represents a certain feature in an area of the original image, and spatial attention is equivalent to telling the network which area of the original image features should be paid attention to, ultimately improving the performance of the entire network.

### 2.3. Our CSConv2d Model

In general, our CSConv2d model is designed by embedding CBAM, to perform attention operations, in the framework of DEEPScreen [19]. The structure of our model is shown in Figure 3, which takes two-dimensional compound images as the input of the network.

In our model, each compound is represented by a 200-by-200-pixel 2-D image displaying the molecular structure generated by RDkit to make the representation unique and ensure the structure standard. Such a 2-D image is entered into the deep convolutional neural networks, which is composed of five convolutional + pooling, a channel attention module and a spatial attention module, and one fully connected layer preceding the output layer, and the DTI prediction was considered as a binary classification problem where the output could either be positive or negative. Each convolutional layer is followed by a ReLU activation function and max pooling layers. The last convolutional layer is flattened and connected to a fully connected layer, followed by the output layer. We use the Softmax activation function in the output layer. The CSConv2d can train a model for each target, and each model is independently optimized to accurately predict interacting small molecule ligands for a unique target protein.

### 2.4. Evaluation Metrics

We select the commonly used evaluation metrics, including Precision, Recall, Accuracy (ACC), F1-score and the Matthews correlation coefficient (MCC). The metrics used in the evaluation of our model are shown in Table 1.
(1)$$\mathrm{Precision}=\frac{\mathrm{TP}}{\mathrm{TP}+\mathrm{FP}}\;\mathrm{Range}\;[0,1]$$
(2)$$\mathrm{Recall}=\frac{\mathrm{TP}}{\mathrm{TP}+\mathrm{FN}}\;\mathrm{Range}\;[0,1]$$
(3)$$\mathrm{ACC}=\frac{\mathrm{TP}+\mathrm{TN}}{\mathrm{TP}+\mathrm{FP}+\mathrm{FN}+\mathrm{TN}}\;\mathrm{Range}\;[0,1]$$
(4)$$F1-\mathrm{score}=\frac{2\times \mathrm{precision}\times \mathrm{recall}}{\mathrm{precision}+\mathrm{recall}}\;\mathrm{Range}\;[0,1]$$
(5)$$\mathrm{MCC}=\frac{TP\times TN-FP\times FN}{\sqrt{\left(TP+FP\right)\times \left(TP+FN\right)\times \left(TN+FP\right)\times \left(TN+FN\right)}}\;\mathrm{Range}[-1,1]$$

TP represents the number of correctly predicted interacting drug-target pairs, and FN represents the number of interacting drug-target pairs that are predicted as noninteracting. TN represents the number of correctly predicted noninteracting drug-target pairs, whereas FP represents the number of noninteracting drug-target pairs that are predicted as interacting.

## 3. Results

### 3.1. Performance of the CSConv2d Model

Data experiments are conducted on ChEMBLv23 datasets. The simulation results show that in predicting protein-ligand binding affinity, our CSConv2d performs better than the DEEPScreen model.

We trained and tested about 800 targets in total. The model is evaluated by ACC, F1-score and MCC. Each target is trained by a separated model, and their ACC, F1-score and MCC indexes are obtained respectively. The average value of the indicators of all the models is taken, as shown in Table 1. The performance of our CSConv2d is better than that of the original model. The ACC value of our model is improved by 2% and the value of the F1-score is improved by 3%. In addition, the MCC value calculated using Equation (5) is 0.67, demonstrating that the DTIs predicted by our model are more accurate than the original model.

We also compared the performance of the same datasets between our model and the original model using a scatter plot of different indexes (shown in Figure 4). As expected, the performance of our model was better than the original model.

### 3.2. Comparison with Different Models

We compare our CSConv2d with DeepConv-DTI proposed in 2019 [20], CPI-Prediction proposed in 2018 [21] and DeepGS proposed in 2020. The comparison results are shown in Table 2. In the comparison models, SMILES and the sequence of the protein were used as the input of the model. We randomly selected a target (Uniprot ID: P00797) trained in our model, then obtained the sequence of the protein and the SMILES of the compound dataset corresponding to the target as the input of the comparison models. In addition, we also added a channel attention mechanism and spatial attention mechanism to the structure of two of the models that were compared with CSConv2d. As shown in Table 2, it is found that the metrics of CSConv2d are higher than those of DeepConv-DTI, CPI-Prediction and DeepGS. In addition, in order to further reflect the performance of our model, we compared the models on a ChEMBL bioactivity benchmark set [22], and the results are shown in Table 3. On the benchmark set, the performance of our model is still the best.

### 3.3. The Robustness

To investigate if CSConv2d could also improve the robustness of the original model, we randomly selected a protein, DLK in complex with inhibitor 2-((6-(3,3-difluoropyrrolidin-1-yl)-4-(1-(oxetan-3-yl)piperidin-4-yl)pyridin-2-yl)amino) isonicotinonitrile (PDB ID: 5ceo) [23], and its unique ligand’s chemical ID was 50D from RCSB PDB [24], as shown in Figure 5. We first extracted the proto-ligand from the protein and redocked it using AutoDock Vina [25]. The RMSD calculated value between the redocked and RX-ligand is 1.3885. This is used as a benchmark. Furthermore, we randomly selected four compounds as a comparison. The results of the molecular docking show that all of the five compound molecules can dock with protein successfully, and the results are shown in Table 4. Simultaneously, interaction diagrams are provided in the Appendix A (Appendix A). In particular, 50D, which is the original ligand of the protein, has the best performance of molecular docking.

According to the results of the prediction, the performance of our model is better than that of the DEEPScreen, which shows that our model has a better robustness. In addition, we also conducted comparative experiments on a series of kinase inhibitors, such as mTOR, VEGFR and JAK. The relevant data and results are shown in the Appendix A (Appendix A). The prediction performance of our model has been significantly improved.

## 4. Conclusions

In this study, we propose a novel deep learning method for drug-target interactions (DTIs) prediction. The proposed model is named CSConv2d, which is improved from the original DEEPScreen model. It has a convolutional block attention module (CBAM) via the use of 2-D structural representations of compounds as input instead of sequential features such as SMILES or molecular fingerprint. Based on the CBAM, we added a channel attention mechanism and spatial attention mechanism after the last convolution layer of the network in order to improve the nonlinear expression of the network. The proposed CSConv2d can train a model for each target, and each model is independently optimized to accurately predict interacting small molecule ligands for a unique target protein. We also try to add them in the first layer of the network, but the effect is not good. We confirmed experimentally that CSConv2d outperformed the original model.

Data experiments are conducted on ChEMBLv23 datasets. The simulation results show that, in predicting protein-ligand binding affinity, our CSConv2d performs better than the original DEEPScreen model as well as some state-of-the-art DTI prediction methods including DeepConv-DTI, CPI-Prediction, CPI-Prediction+CS, DeepGS and DeepGS+CS. In practice, the docking results of protein (PDB ID: 5ceo) and ligand (Chemical ID: 50D) and a series of kinase inhibitors, such as mTOR, VEGFR and JAK, are operated to verify the robustness.

## Figures and Tables

**Figure 1 biomolecules-11-00643-f001:**
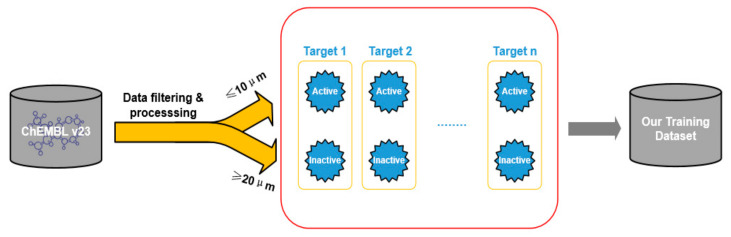
Data filtering and processing to create the training dataset of each target protein.

**Figure 2 biomolecules-11-00643-f002:**
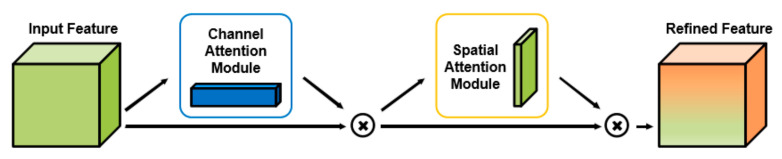
The overview of CBAM.

**Figure 3 biomolecules-11-00643-f003:**
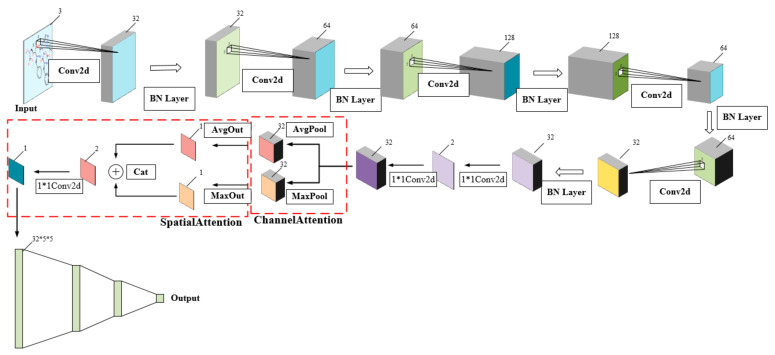
The structure of our model. It has five layers of convolution, channel attention and spatial attention blocks, and four layers of dense layer. CBAM blocks can not only effectively enhance the performance but are also computationally lightweight and impose only a slight increase in model complexity and computational burden.

**Figure 4 biomolecules-11-00643-f004:**
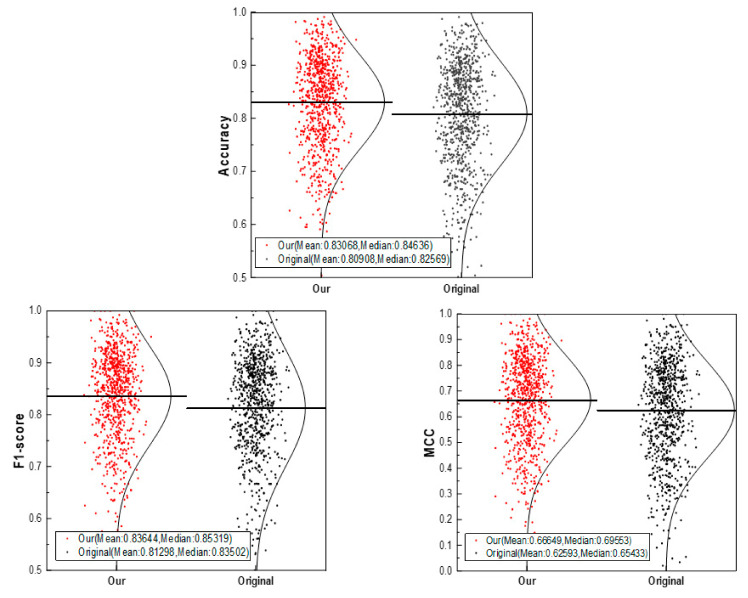
Performance comparison diagram of CSConv2d vs. original model.

**Figure 5 biomolecules-11-00643-f005:**
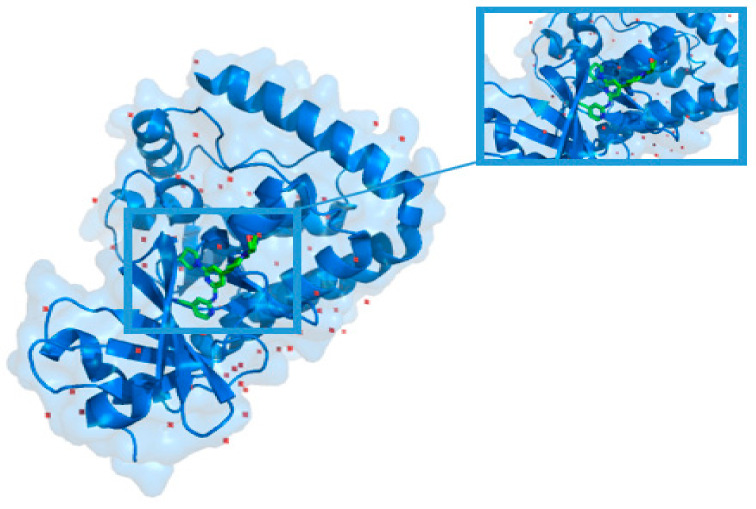
The complex of protein (PDB ID: 5ceo) and ligand (Chemical ID: 50D).

**Table 1 biomolecules-11-00643-t001:** Comparison list of the mean values of accuracy (ACC), F1-score and Matthews correlation coefficient (MCC) between our model and DEEPScreen in a test set.

Model	ACC	F1-Score	MCC
CSConv2d (our)	0.83	0.84	0.67
DEEPScreen (Original)	0.81	0.81	0.63

**Table 2 biomolecules-11-00643-t002:** Comparison list of the accuracy (ACC), F1-score and Matthews correlation coefficient (MCC) of different models in a test set.

Model	ACC	F1-Score	MCC
DeepConv-DTI	0.79	0.80	0.64
CPI-Prediction CPI-Prediction+CS DeepGS DeepGS+CS CSConv2d	0.76 0.80 0.83 0.85 0.87	0.81 0.83 0.85 0.86 0.89	0.63 0.67 0.66 0.70 0.72

**Table 3 biomolecules-11-00643-t003:** Comparison list of the Matthews correlation coefficient (MCC) of different models in the ChEMBL Bioactivity Benchmark Set.

Dataset	Model	MCC
ChEMBL Bioactivity Benchmark Set [23]	Feed-forward DNN PCM	0.38
	CPI-Prediction+CS	0.42
	DeepGS+CS	0.47
	DEEPScreen	0.47
	CSConv2d	0.57

**Table 4 biomolecules-11-00643-t004:** Results of the molecular docking and the predicted result by CSConv2d and the original model.

Compound ID	Affinity (kcal/mol)	Pred (CSConv2d)	Pred (DEEPScreen)	Label
50D	−9.2	1	0	1
CHEMBL3731242	−8.7	1	1	1
CHEMBL3729274	−8.1	1	1	1
CHEMBL3355005	−7.9	1	1	1
CHEMBL3727745	−7.9	1	0	1
CHEMBL1242663	−7.5	0	0	0
CHEMBL1767275	−7.4	0	1	0
CHEMBL598911	−6.8	1	0	0
CHEMBL206659	−6.2	0	1	0
CHEMBL25829	−4.7	0	0	0

## Data Availability

The raw dataset is available for download at https://doi.org/10.4121/uuid:547e8014-d662-4852-9840-c1ef065d03ef (accessed on 31 July 2017).

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
