# Peer review of "CSConv2d: A 2-D Structural Convolution Neural Network with a Channel and Spatial Attention Mechanism for Protein-Ligand Binding Affinity Prediction"

_biomolecules, 2021, doi:10.3390/biom11050643_

Round 1
Reviewer 1 Report
In this manuscript (”CSConv2d: A Novel 2-D Structural Convolution Neural Network with Channel and Spatial Attention Mechanism for Protein-Ligand Binding Affinity Prediction” is described a novel deep learning method for drug-target interactions (DTIs) prediction. The study is of interest to the scientific community, but there are few points to address.
The first main recommendation is to reorganize and rewrite some of the manuscript sections. The authors should follow a clear and intelligible way of presenting their work. From my point of view, in this submitted version, the manuscript does not present the expected impact. Some sections are very difficult to understand because they are not sufficiently explained.
By addressing all the recommendations, the quality of the manuscript will be improved and it will certainly be able to meet the requirements of the journal publication.
- Lines 85 and 86 (bioactivity values ≤10M and ≥20M) and in Figure 1 (≤10μm and ≥20μm). Please explain.
- Line 130 Equation (5)?. In subsection “2.4 Evaluation Metrics”, equations are not numbered. I suggest to number: ACC (1), F1-score (2), and MCC (3) which are used in the evaluation model (Table 2), and Recall and Precision as explanations for equation 2.
-Lines 140 and 213 “Experimental results…” From my point of view it should be replaced with the “Simulation results…”
- Line 143. In the caption of Table 1, please specify that there are presented the average values of the indicators
- Could you please include in the results section statistical data that support better all sentences?
E.g. Line 142 “….original model” Which is the original model? Please explain
- Line 173- the subsection “3.3 The robustness” Why did you select only the protein DLK (PDB ID: 5ceo)? There is no information in this regard.
- What is the RMSD calculated value between re-docked and RX-ligand?
- Line 178 “As well, we randomly selected four compounds as a comparison.” From my point of view, four randomly selected compounds are too few to extract relevant conclusions.
- Line 190 – Please reorganize Figure 6, because in part C and D of the figure the information is the same as in part A’s blue square. Figure 6 is not explained/mentioned in the main text of the manuscript
- Line 230 - [J] is it necessary?
- It is very important to add supplementary material to explain/complete some unclear/missing information in the manuscript
The significant outcomes provided by the manuscript, if written in a clear and easy-to-follow manner, could be a real win for researchers interested in this field. It is strongly recommended that the authors read the manuscript once again after having made all the indicated corrections and make it more accessible for readers.
Reviewer 2 Report
The manuscript "CSConv2d: A Novel 2-D Structural Convolution Neural Network with Channel and Spatial Attention Mechanism for Protein-Ligand Binding Affinity Prediction" by Xun Wang et al. presents an improvement of the DEEPScreen model, namely CSConv2d, for prediction of drug-target interactions (DTIs).
Overall the work is interesting and I suggest it's publication to Biomolecules, after the authors address the following issues:
- Given that the use of 2D structural representations of compounds as input instead of molecular fingerprints is not a novel method, the authors should avoid the use of 'novel'. Their improvement is based on the use of a convolutional block attention module (CBAM) to enhance the non-linear expression ability of the network.
- Considering that There are critical points and risks in constructing training and test datasets, methods should be described in more detail. In particular, the authors should give exact numbers of positive and negative samples, as well as present the filtering stages in more detail.
- Similartly, the authors should have addressed the issue of negative selection bias in their dataset (see for example ref. 19)
- In the comparison analysis, the authors should present the results of CSConv2d method using at least one common benchmark, e.g. ChEMBL temporal-split dataset or the maximum unbiased validation (MUV) dataset.
- The robustness (par. 3.3) of CSConv2d should be demonstrated for a series of protein-ligand complexes (at least 3) and not just a single case (e.g. for a series of kinase inhibitors).
- Minor issues: Panels A and B in Fig. 6 are streched inproportionally and overall fonts are illegible (this figure could be supplied as Supplementary Material).
Round 2
Reviewer 1 Report
The authors have satisfactorily responded to all my questions and made the necessary changes to the manuscript. I have no further comment on the revised version of the manuscript “CSConv2d: A Novel 2-D Structural Convolution Neural Network with Channel and Spatial Attention Mechanism for Protein-Ligand Binding Affinity Prediction”. It looks ready for publication.